# BAF60a deficiency uncouples chromatin accessibility and cold sensitivity from white fat browning

Tongyu Liu[1,2,6], Lin Mi[1,6], Jing Xiong[1], Peter Orchard[2,3], Qi Yu[1], Lei Yu[1], Xu-Yun Zhao[1], Zhuo-Xian Meng[4,5], Stephen C. J. Parker[2,3], Jiandie D. Lin[1✉] & Siming Li[1✉]

Brown and beige fat share a remarkably similar transcriptional program that supports fuel oxidation and thermogenesis. The chromatin-remodeling machinery that governs genome accessibility and renders adipocytes poised for thermogenic activation remains elusive. Here we show that BAF60a, a subunit of the SWI/SNF chromatin-remodeling complexes, serves an indispensable role in cold-induced thermogenesis in brown fat. BAF60a maintains chromatin accessibility at PPARγ and EBF2 binding sites for key thermogenic genes. Surprisingly, fat-specific BAF60a inactivation triggers more pronounced cold-induced browning of inguinal white adipose tissue that is linked to induction of MC2R, a receptor for the pituitary hormone ACTH. Elevated MC2R expression sensitizes adipocytes and BAF60a-deficient adipose tissue to thermogenic activation in response to ACTH stimulation. These observations reveal an unexpected dichotomous role of BAF60a-mediated chromatin remodeling in transcriptional control of brown and beige gene programs and illustrate a pituitary-adipose signaling axis in the control of thermogenesis.

[1] Life Sciences Institute and Department of Cell and Developmental Biology, University of Michigan, Ann Arbor, MI 48109, USA. [2] Department of Computational Medicine and Bioinformatics, University of Michigan, Ann Arbor, MI 48109, USA. [3] Department of Human Genetics, University of Michigan, Ann Arbor, MI 48109, USA. [4] Department of Pathology and Pathophysiology, Key Laboratory of Disease Proteomics of Zhejiang Province, Hangzhou, Zhejiang 310058, China. [5] Chronic Disease Research Institute of School of Public Health, School of Medicine, Zhejiang University, Hangzhou, Zhejiang 310058, China. [6] These authors contributed equally: Tongyu Liu, Lin Mi. ✉email: jdlin@umich.edu; simingli@umich.edu

Adipose tissue thermogenesis defends against cold stress and contributes to nutrient homeostasis and energy balance. Recent studies have identified beige (or brite) adipocytes as an important thermogenic cell type in rodents and humans[1–4]. Brown and beige fat share a remarkably similar transcriptional program underlying fuel oxidation and engage several biochemical processes to convert chemical energy into heat, including uncoupled respiration via uncoupling protein 1 (UCP1) and UCP1-independent thermogenesis via futile substrate cycling. Beyond thermogenesis, brown and beige fat also release endocrine hormones such as Neuregulin 4, which act on other tissues to influence metabolic physiology and disease pathogenesis[5–7]. While the clinical and therapeutic importance of adipose tissue thermogenesis remains to be fully established, brown and beige fat mass and activity have generally been associated with desirable metabolic parameters[8]. Importantly, cold-induced and pharmacological activation of adipose thermogenesis led to increased glucose utilization and improved insulin sensitivity in humans[9,10]. In rodents, genetic ablation of brown fat exacerbates diet-induced weight gain and metabolic disorders[11], whereas its activation results in increased energy expenditure, reduced adiposity, improved insulin sensitivity, and plasma lipid profiles[12–14]. These observations illustrate the conserved nature of the beneficial effects of thermogenic activation on systemic metabolism and physiology.

A network of transcriptional regulators have been identified to control brown and beige adipogenesis and thermogenesis, including transcription factors (e.g., PPARγ, EBF2, C/EBPβ, ZBTB7B), cofactors (PRDM16, PGC-1α, HDAC3), histone-modifying epigenetic regulators (EHMT1, JMJD1A, KDM4B, LSD1, HDAC), and long noncoding RNAs (Blnc1, lncBATE)[15,16]. Many of these factors form transcriptional complexes through physical interaction and are recruited to the regulatory regions of the genes underlying mitochondrial oxidation and thermogenesis in response to thermogenic stimuli. The mammalian SWI/SNF chromatin-remodeling complexes exist in three distinct assemblies: canonical BRG1-associated factor (cBAF), polybromo-associated BAF, and non-canonical BAF (ncBAF) complexes[17–20]. These heterogeneous molecular machines regulate diverse biological processes by modulating nucleosome structure, chromatin accessibility, and gene transcription. Recent works have shown that brown adipocyte gene expressions are regulated by EBF2 recruitment of BRG1- and JMJD1A-associated SWI/SNF complexes[21,22]. Despite these, the role of SWI/SNF-mediated chromatin regulation in cold-induced activation of the brown and beige gene programs remains largely unknown. In this study, we generated mice harboring fat-specific inactivation of BAF60a, a molecular link between the SWI/SNF machinery and chromatin. Our work uncovers a surprisingly dichotomous role of BAF60a in governing the thermogenic gene programs and reveals a pituitary-adipose signaling pathway that drives cold-induced browning.

## Results

### The BAF60a transcriptional program is indispensable for cold-induced thermogenesis.
The mammalian SWI/SNF complexes contain several core subunits including BAF60 that undergo orderly assembly into functional molecular machines[19]. Previous studies have demonstrated that the BAF60 subunit plays a uniquely important role in recruiting the SWI/SNF chromatin-remodeling complexes to target chromatin, leading to cell type-specific and context-dependent transcriptional response[23]. BAF60a mediates induction of hepatic genes involved in fatty acid β-oxidation and bile acid synthesis[24,25], whereas BAF60c serves as a core component of myocyte glucose sensing and orchestrates myogenic gene expression and glycolytic muscle metabolism[26–28].

As BAF60a is abundantly expressed in brown fat, we postulated that this factor may provide an important link between SWI/SNF-mediated chromatin remodeling and thermogenesis. To test this hypothesis, we inactivated BAF60a specifically in adipose tissue and explored its role in gating chromatin accessibility during thermogenic response.

Adipocyte-specific BAF60a knockout (AKO) mice exhibited normal postnatal growth and appeared indistinguishable from flox/flox littermate control (flox). Following cold exposure, AKO mice exhibited heightened cold sensitivity and failed to maintain core body temperature; this thermogenic defect was associated with elevated plasma levels of triglycerides, ketone body (β-hydroxybutyrate), and non-esterified fatty acids (NEFAs) compared to control (Fig. 1a, b). Histological staining revealed that, while interscapular BAT appeared similar in two groups when mice were kept at ambient temperature, cold-induced mobilization of lipid storage failed to materialize, as indicated by the presence of larger lipid droplets in BAF60a-deficient brown adipocytes (Fig. 1c). Lipolytic function of epididymal white adipose tissue (eWAT) appeared to be enhanced by BAF60a inactivation under basal and stimulated conditions (Supplementary Fig. 1a). Quantitative PCR (qPCR) analysis indicated that mRNA expression of genes involved in uncoupling and thermogenesis (Ucp1, Dio2), fatty acid β-oxidation (Cpt1b, Acaa2, Mcad, Acadl, Elovl3), and mitochondrial oxidative metabolism (Cox7a1, Cox8b) was significantly lower in AKO mouse brown fat, whereas BAF60b and BAF60c mRNA levels remained largely unaltered by BAF60a inactivation in BAT, inguinal WAT (iWAT), and eWAT from AKO mice (Fig. 1d and Supplementary Fig. 1b). Accordingly, BAF60a inactivation markedly decreased UCP1 protein levels and disrupted normal mitochondrial morphology in brown adipocytes lacking BAF60a, as revealed by the appearance of dilated mitochondrial cristae (Fig. 1e, f). We surveyed mRNA expression of several known regulators of adipose tissue thermogenesis. While Ebf2, Zbtb7b, and Prdm16 mRNA levels remained comparable in two groups, mRNA expression of PGC-1α, Blnc1, and PPARγ2 was significantly lower in AKO brown fat (Fig. 1d).

Cold exposure stimulates catecholamine release by the sympathetic nerve fibers in adipose tissue, leading to adrenergic receptor activation and thermogenesis. To directly assess whether thermogenic response to adrenergic stimulation is impaired in AKO mice, we performed thermal imaging on control and AKO mice following a single intraperitoneal (i.p.) injection of CL-316,243, a β3-selective adrenergic agonist and potent inducer of brown fat thermogenesis. As expected, CL-316,243 treatment increased surface body temperature in control flox mice (Fig. 1g, h). In contrast, AKO mice failed to mount a robust thermogenic response following CL-316,243 treatment. In fact, surface body temperature remained similar to saline-treated mice. These results illustrate an indispensable role for BAF60a-mediated chromatin remodeling during cold-induced thermogenesis.

### BAF60a governs chromatin accessibility in brown fat.
BAF60a mediates the recruitment of the SWI/SNF complexes to selective genomic loci, thereby altering local chromatin structure and gene transcription[23,29]. We next performed assay for transposase-accessible chromatin using sequencing (ATAC-seq) on nuclei isolated from control and AKO mouse brown fat to delineate the role of BAF60a in setting chromatin accessibility. ATAC-seq provides a powerful tool to reveal the landscape of chromatin accessibility across the entire genome[30]. We identified a total of 47,205 ATAC-seq peaks (representing open chromatin) in brown fat, of which 265 exhibiting significantly reduced accessibility in response to BAF60a inactivation (Fig. 2a). Interestingly, 72 peaks

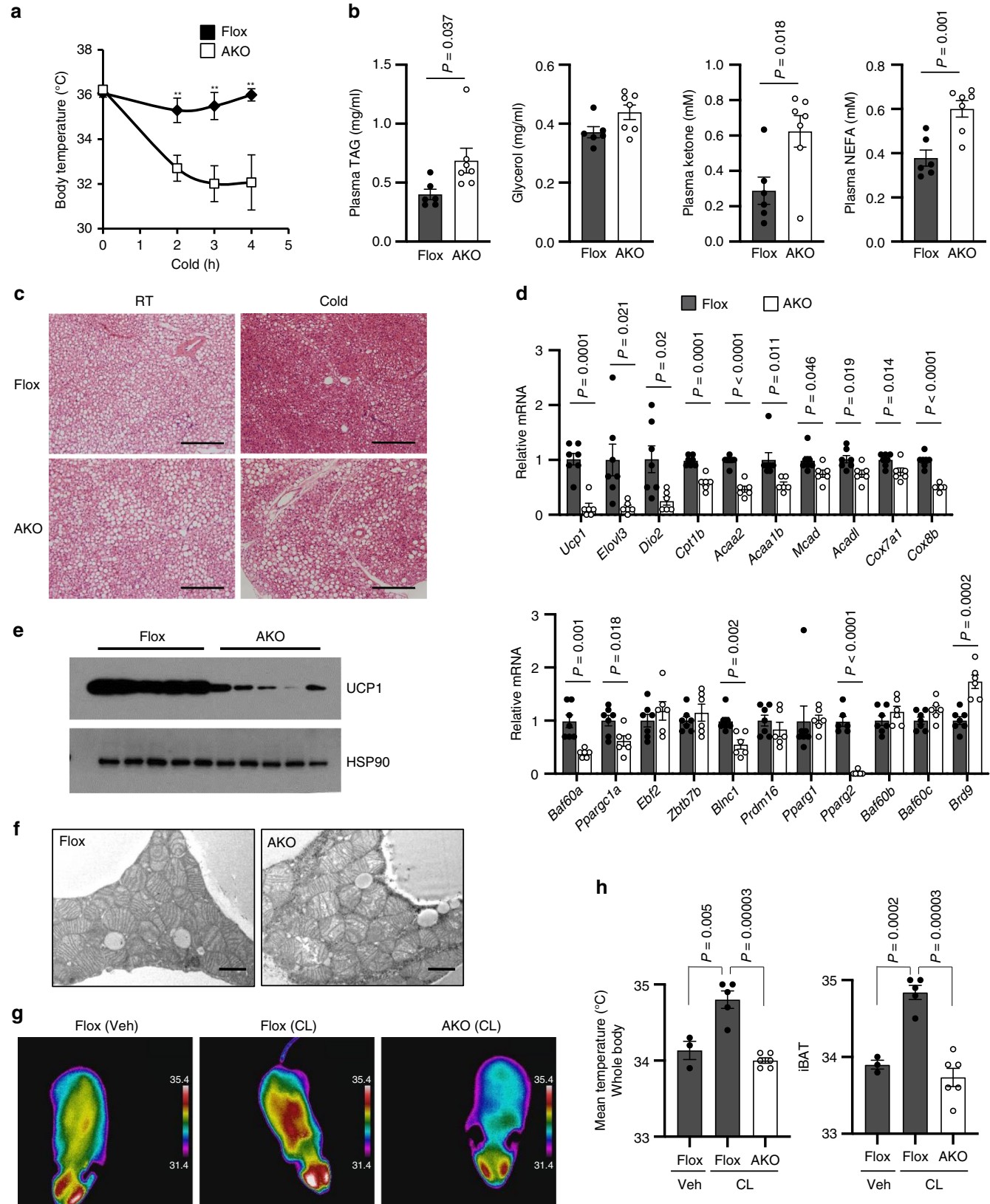

exhibited increased accessibility in AKO mouse brown fat. We analyzed the effects of BAF60a deficiency on a set of brown fat-enriched genes and observed reduced accessibility at ATAC-seq peaks and mRNA expression for many of these genes, including *Pparγ*, *Ifi27l2a*, *Cfd*, *Dgat2*, and *Pck1* (Fig. 2a and Supplementary Fig. 1c). Seven ATAC-seq peaks were detected across the Ucp1

locus, all of which exhibited reduced chromatin accessibility (Fig. 2b), consistent with significantly lower *Ucp1* gene expression in AKO mouse brown fat.

To determine whether BAF60a preferentially affects chromatin accessibility at certain transcription factor binding sites, we performed transcription factor motif-enrichment analysis on

**Fig. 1 BAF60a is indispensable for cold-induced brown fat thermogenesis. a** Rectal body temperature in flox ($n = 15$, filled) and AKO ($n = 20$, open) mice during cold exposure at 4 °C. Data were pooled from two experiments and analyzed using two-way ANOVA with Bonferroni post hoc test. Data represent mean ± s.e.m. Interaction $P$ value = 0.004; **$P < 0.01$ for each time point; flox vs. AKO. **b** Plasma metabolite concentrations following cold exposure in flox ($n = 6$, filled) and AKO ($n = 7$, open) mice. **c** Histology of brown fat from mice kept at ambient room temperature (RT) or after cold exposure for 6 h at 10 °C (scale bar = 200 μm; $n = 3$ per group). **d** qPCR analysis of BAT gene expression in flox ($n = 7$, filled) and AKO ($n = 6$, open) mice kept at ambient room temperature. **e** Immunoblots of brown fat lysates. **f** Transmission electron micrograph of BAT sections (scale bar = 1 μm, $n = 3$ per group). **g** Representative infrared images of flox and AKO mice 3 h after i.p. injection of saline (Veh) or CL-316,243 (CL). **h** Quantitation of mean body surface temperature on the back (whole body) and the area above interscapular BAT (iBAT) (saline flox, $n = 3$, filled; CL flox, $n = 5$, filled; CL AKO, $n = 6$, open). Data in **b**, **d**, **h** represent mean ± s.e.m., two-tailed unpaired Student's $t$ test. Source data are provided as a Source Data file.

ATAC-seq peaks that were decreased by BAF60a inactivation. We observed that the binding sites for several nuclear receptors, including PPAR, NUR77, RXR, and ERRβ, were highly enriched in BAF60a down-regulated peaks (Fig. 2c). We integrated our ATAC-seq data with previous chromatin immunoprecipitation sequencing (ChIP-seq) analyses of genome-wide PPARγ and EBF2 occupancy in brown fat[21]. As shown in Fig. 2d, 16,868 of 23,232 PPARγ and 6610 of 7584 EBF2 ChIP peaks were found within ATAC-seq peaks that we observed in brown fat. Interestingly, a large fraction of BAF60a-dependent ATAC peaks contained PPARγ (247/337) and EBF2 (92/337) binding sites. Co-immunoprecipitation (co-IP) studies indicated that BAF60a physically interacts with PPARγ (Supplementary Fig. 2a). Transcription factor footprint analysis of the ATAC-seq peaks indicated that chromatin regions near PPARγ binding sites exhibited reduced accessibility (Fig. 2e). Accordingly, ChIP analysis revealed diminished PPARγ binding at the Ucp1 locus (Supplementary Fig. 2b). In contrast, occupancy for CCCTC-binding factor (CTCF), a chromatin-binding factor that defines the boundary of chromatin domains, remained largely unaffected by inactivation of BAF60a in brown adipocytes. These results provide a mechanistic link between BAF60a-mediated chromatin remodeling and the global landscape of genome accessibility in brown fat.

**BRD9 interacts with BAF60a and is required for thermogenic gene expression.** To establish a cell-autonomous role of BAF60a in the control of thermogenic gene program, we immortalized brown preadipocytes from flox/flox neonatal pups and ablated BAF60a following transduction of a retroviral vector expressing Cre recombinase. As expected, expression of *Ucp1*, *Cidea*, and *Fabp4* was markedly induced in control adipocytes during adipogenesis (Fig. 3a, b and Supplementary Fig. 3a). In contrast, the induction of these thermogenic genes was greatly diminished in adipocytes lacking BAF60a, which exhibited reduced lipid accumulation following differentiation (Supplementary Fig. 3b). Further, adrenergic stimulation of thermogenic gene expression was greatly diminished in BAF60a-deficient adipocytes (Fig. 3c). MitoTracker staining indicated that mitochondrial content was lower in adipocytes lacking BAF60a (Fig. 3d). Consistently, basal oxygen consumption and total respiratory capacity of differentiated adipocytes were significantly lower following BAF60a inactivation (Fig. 3e). As such, BAF60a is required for transcriptional activation of thermogenic genes in a cell-autonomous manner.

BAF60a is a core subunit of the molecularly heterogeneous BAF chromatin-remodeling complexes. Recent studies have identified bromodomain-containing protein 9 (BRD9) as a subunit of ncBAF in mouse embryonic stem cells and cancer cells[17,20,31,32]. We performed co-IP assay to explore whether BAF60a is present in BRD9-containing ncBAF in brown adipocytes. As shown in Fig. 3f, BAF60a physically associated with BRD9 in addition to BRG1 and BAF47, suggesting that BAF60a is present in both cBAF and ncBAF complexes. These

findings raised the possibility that BRD9-containing ncBAF may play an important role in the regulation of thermogenic gene expression. To explore this, we subjected brown preadipocytes to differentiation protocol without or with dBRD9, a BRD9-specific proteolysis targeting chimera compound[33]. As expected, dBRD9 markedly diminished endogenous BRD9 protein levels in treated adipocytes (Fig. 3g). While protein levels of FASN (fatty acid synthase) and PPARγ remained largely unaffected by dBRD9, UCP1 mRNA and protein expression was greatly reduced in differentiated brown adipocytes (Fig. 3g, h). We confirmed these results using a specific chemical inhibitor of BRD9 (I-BRD9)[34]. Both I-BRD9 and dBRD9 diminished the induction of *Ucp1* mRNA and protein expression in response to adrenergic stimulation (Fig. 3i, j). These results demonstrate that BRD9 is required for induction of thermogenic gene expression during brown adipocyte differentiation and in response to adrenergic stimulation.

**Inactivation of BAF60a enhances beige fat formation.** Chronic cold exposure is known to stimulate browning of white fat, particularly the iWAT. Beige adipocytes within iWAT contribute to thermogenesis via UCP1-dependent and UCP1-independent mechanisms[4]. To determine whether BAF60a is required for beige fat formation, we subjected flox and AKO mice to cold acclimation. Unlike acute cold stress, chronic cold exposure reduced brown fat lipid content in both groups (Fig. 4a). Despite this, mRNA expression of *Ucp1*, *Cpt1b*, *Elovl3*, and *Cox8b* and UCP1 protein levels were significantly lower in BAF60a-deficient brown fat (Fig. 4b, c). To our surprise, we observed more pronounced cold-induced browning in iWAT from AKO mice, as illustrated by more abundant UCP1-positive beige adipocytes (Fig. 4c, d). Gene expression analysis indicated that BAF60a was efficiently deleted in adipocytes along with markedly reduced expression of PPARγ2 (Supplementary Fig. 4a). Accordingly, mRNA expression of a large number of genes involved in mitochondrial fuel oxidation and thermogenesis was more highly induced in AKO mouse iWAT than control, including *Ucp1*, *Elovl3*, *Cidea*, *Cpt1b*, and *Pparα* (Fig. 4e). Dio2 mRNA levels were comparable between two groups. Microarray transcriptional profiling of iWAT revealed a cluster of genes exhibiting more robust induction in AKO mice following cold acclimation, many of which are involved in mitochondrial fuel oxidation and thermogenesis (Fig. 4f). The expression of genes involved in creatine metabolism (*Ckmt1*, *Ckmt2*, *Gatm*, *Gamt*, *Slc6a8*) remained largely unchanged (Supplementary Fig. 4b). While *Serca2a* was upregulated in AKO mice during chronic cold, the expression of *Serca2b* and *Ryr2*, two major players in calcium cycling in response to cold challenge[35], were not affected by BAF60a inactivation (Supplementary Fig. 4b).

Recent work has uncovered a unique population of beige adipocytes characterized by elevated glycolytic activity[36]. Interestingly, mRNA expression of a number of glycolytic genes, including *Hk2*, *Eno1*, *Pfkp*, *Pkm2*, and *Ldha*, was significantly higher in iWAT from AKO mice following cold acclimation than

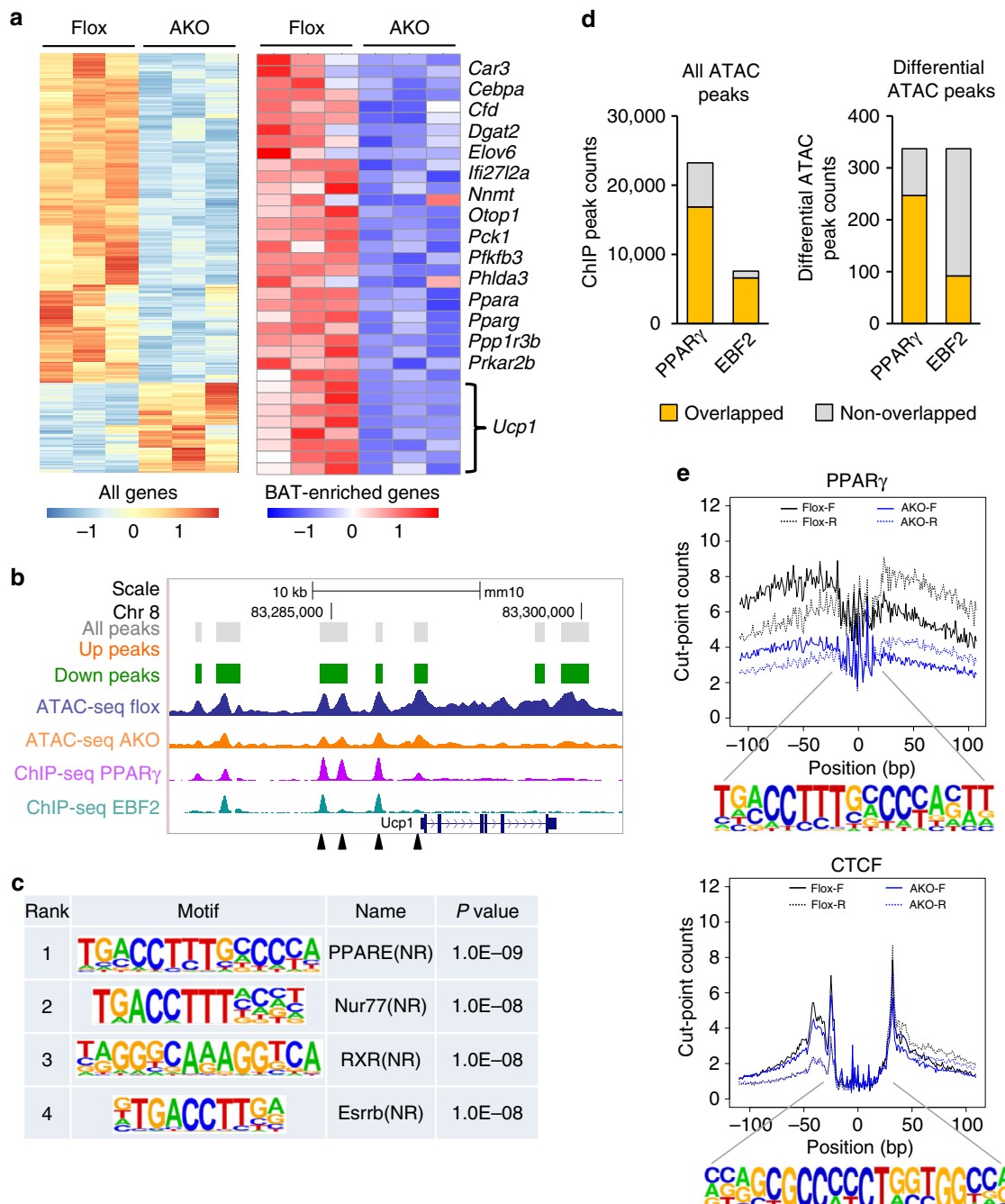

**Fig. 2 Ablation of BAF60a disrupts chromatin accessibility in brown fat.** ATAC-seq was performed on brown fat nuclei isolated from flox ($n = 3$) and AKO ($n = 3$) mice. **a** Heatmap representation of all differential peaks (left) or differential peaks near BAT-enriched genes (right). **b** UCSC genome browser view of chromatin structure at the Ucp1 locus. Mapped ATAC-seq peaks correspond to all peaks and those up- or down-regulated following BAF60a inactivation. ATAC-seq traces for Flox and AKO brown fat (middle) and the ChIP-seq peaks for PPARγ and EBF2 (bottom) were shown. Arrowheads indicate the locations for PPARγ ChIP qPCR primers. **c** Motif-enrichment analysis of the down-regulated peaks in AKO brown fat. **d** Overlaps of PPARγ and EBF2 ChIP binding peaks within all ATAC-seq peaks (left) or differential ATAC-seq peaks (right). **e** Footprint aggregate plots at putative PPARγ and CTCF binding sites. Source data are provided as a Source Data file.

the control group (Fig. 4g). We next performed ATAC-seq to examine how BAF60a inactivation alters chromatin accessibility in iWAT following cold acclimation. We identified a total of 39,558 peaks, 29,925 of which (75.7%) overlapped with peaks detected in BAT (Fig. 5a), indicating that brown and beige fat share remarkably similar features in their chromatin accessibility landscape. Motif-enrichment analysis revealed that the top 12 BAT-enriched motifs were also enriched among peaks detected in iWAT, including binding sites for C/EBP, CTCF, HLF, BORIS,

EBF, PPAR, NF1, and SP1 (Fig. 5b). While ATAC-seq peaks for *Ifi27l2a* and *Car3* were diminished by BAF60a inactivation in both BAT and iWAT, the peaks near *Ucp1* and *Cidea* genes exhibited opposite regulation by BAF60a inactivation (Fig. 5c). Motif analysis indicated that the upregulated peaks in iWAT were enriched for EBF, C/EBP, and PPAR binding sites (Fig. 5d). Together, these observations illustrate a surprisingly dichotomous role of BAF60a in the control of thermogenic gene programs in brown and beige adipocytes. In this case, BAF60a is absolutely

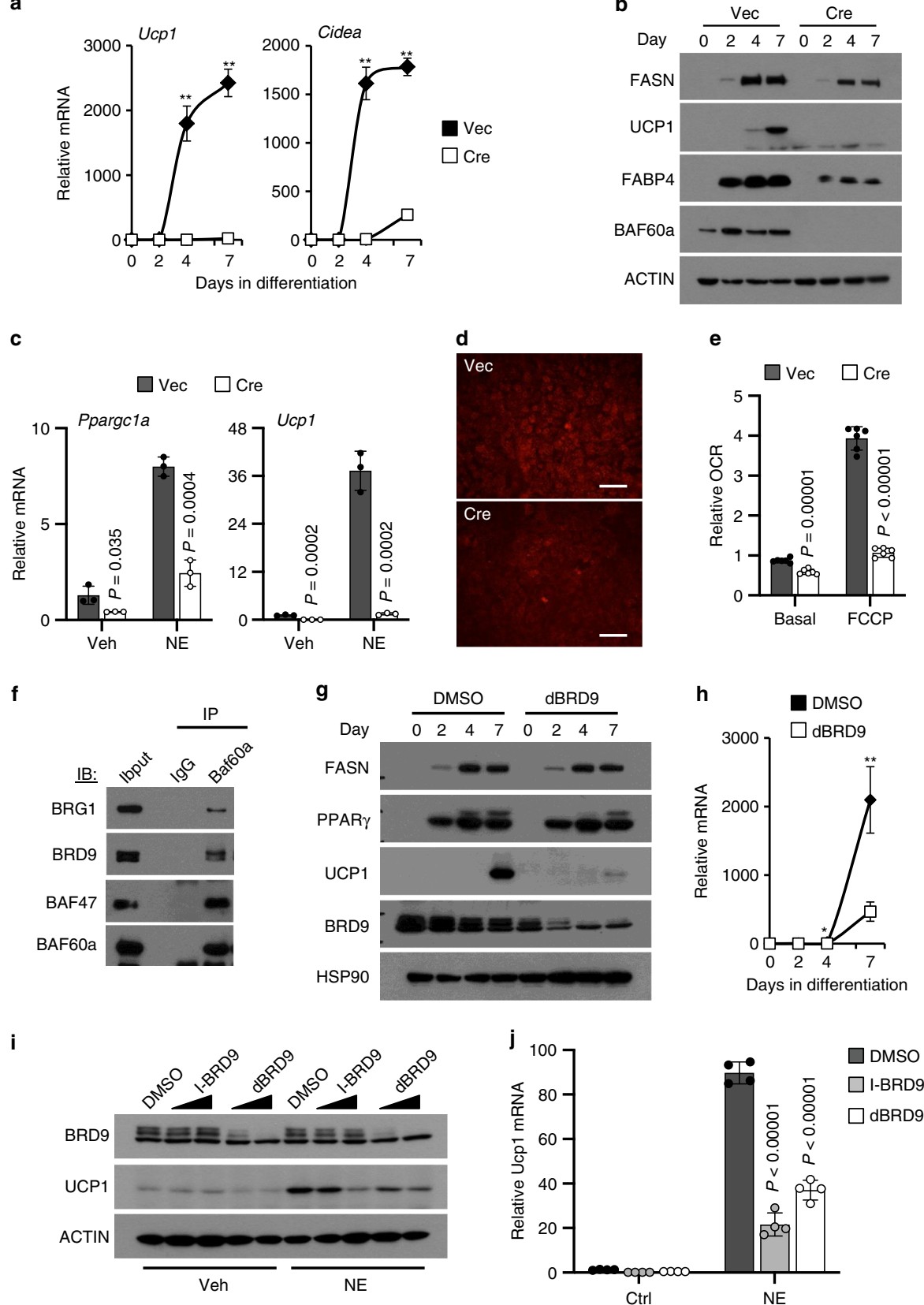

**Fig. 3 Deletion of BAF60a impairs brown adipocyte differentiation. a** qPCR analysis of gene expression during brown adipocyte differentiation following transduction of vector or Cre retrovirus ($n = 3$ per group). **b** Immunoblots of brown adipocyte lysates during differentiation as in **a**. **c** qPCR analysis of differentiated brown adipocyte gene expression following 5 h of treatment with vehicle (Veh, $n = 3$, filled) or norepinephrine (NE, $n = 3$, open). **d** MitoTracker staining (scale bar = 100 μm; $n = 3$ per group). **e** Oxygen consumption rate (OCR) before and after FCCP addition ($n = 6$ per group). Data in **a**, **c**, **e** represent mean ± s.d. **$P < 0.01$, Vec vs. Cre; two-tailed unpaired Student's $t$ test. **f** Immunoblots of BAF60a-associated proteins in brown adipocytes (representative of three experiments). **g** Immunoblots of brown adipocyte lysates during differentiation in the presence of DMSO or 250 nM dBRD9 ($n = 2$ per group). **h** qPCR analysis of $Ucp1$ expression ($n = 3$ per group). **i** Immunoblots of brown adipocyte lysates. Differentiated adipocytes were treated with Veh or NE in the presence of DMSO, I-BRD9 (1 or 10 μM), or dBRD9 (100 or 500 nM) ($n = 2$ per group). **j** qPCR analysis of $Ucp1$ expression in brown adipocytes treated with Veh or NE in the presence of DMSO, 10 μM I-BRD9 or 500 nM dBRD9 ($n = 4$ per group). Data in **h**, **j** represent mean ± s.d. *$P < 0.05$, **$P < 0.01$, DMSO vs. I-BRD9 or dBRD9; two-tailed unpaired Student's $t$ test. Source data are provided as a Source Data file.

required for brown fat thermogenesis and defense against acute cold stress, yet its inactivation appears to sensitize beige fat formation in response to chronic cold stimulation.

**BAF60a deficiency engages ACTH/MC2R signaling axis to induce browning.** Secreted factors play an important role in driving beige fat formation in response to physiological and environmental stimuli. Adrenergic stimulation as a result of increased sympathetic outflow to adipose tissue provides a potent signal that promotes white fat browning. Further, a number of local and endocrine factors have been identified to mediate beige fat induction under different physiological conditions[3]. To explore whether BAF60a inactivation may activate a dormant signaling pathway to enhance browning, we focused our analysis on membrane receptors that exhibit abundant expression in iWAT. Analysis of our microarray dataset revealed that mRNA expression of $Mc2r$, $Oxtr$, and $Kiss1r$, G-protein coupled receptors responding to pituitary hormones, was markedly elevated in iWAT from cold-acclimated AKO mice compared to control (Fig. 6a). In contrast, mRNA levels of other pituitary hormone receptors, including $Avpr1a$, $Ghr$, $Tshr$, and $Prlr$, remained largely unaltered by BAF60a inactivation. Similarly, mRNA expression of $Adrb3$ and $Chrna2$, which mediate the response to nor-epinephrine (NE) and acetylcholine[37], respectively, was comparable in two groups. We next examined the effects of ligand activation of $Mc2r$, $Oxtr$, and $Kiss1r$ on thermogenic genes in adipocytes differentiated from primary stromal vascular cells isolated from iWAT. As expected, both isoproterenol (Iso) and CL-316,243 strongly induced mRNA expression of $Ucp1$ and PGC-1α (Fig. 6b). While KISS1 and OXT (oxytocin), physiological ligands for KISS1R and OXTR, respectively, had modest effects on $Ucp1$ and PGC-1α expression, activation of MC2R by its ligand adrenocorticotropic hormone (ACTH) elicited a strong induction of $Ucp1$ and PGC-1α. Cell fractionation studies indicated that $Mc2r$ induction in iWAT occurred in BAF60a-deficient adipocytes, but not stromal vascular cells (Fig. 6c and Supplementary Fig. 4a). Further, this increase in $Mc2r$ expression was observed in iWAT under thermoneutral condition and in BAT and eWAT from BAF60a AKO mice (Supplementary Fig. 5a, b). We noted the presence of two major ATAC-seq peaks at the proximal promoter and first intron of $Mc2r$ that coincide with PPARγ binding sites (Fig. 6d). Accordingly, rosiglitazone significantly increased mRNA expression of $Mc2r$ in differentiated beige adipocytes (Fig. 6e). We next examined whether altered local release of catecholamines and ACTH levels in circulation may account for enhanced browning in AKO mice. To our surprise, iWAT epinephrine and norepinephrine (NE) and plasma ACTH concentrations were similar in control and AKO mice (Fig. 6f).

ACTH stimulates cAMP signaling and PKA activation and has been shown to regulate thermogenic gene expression[38,39]. To directly assess whether increased MC2R expression may sensitize adipocyte response to ACTH, we performed treatments in

differentiated beige adipocytes expressing GFP or MC2R with different concentrations of ACTH. As shown in Fig. 7a, MC2R overexpression markedly augments the induction of $Ucp1$ and $Elovl3$ mRNA expression in response to ACTH. Consistently, ACTH-induced PKA activation, as revealed by phosphorylation of PKA substrates, was more pronounced in adipocytes over-expressing MC2R (Fig. 7b). RNA interference knockdown of MC2R in beige adipocytes impaired activation of PKA signaling by ACTH, but not Iso (Fig. 7d). Induction of $Ucp1$ and $Elovl3$ mRNA expression in response to ACTH was diminished by knockdown of MC2R (Fig. 7c). These results illustrate that MC2R expression levels modulate hormonal responsiveness of beige adipocytes to ACTH. To further assess whether the induction of $Mc2r$ expression sensitize ACTH response in vivo, we cultured adipose tissue explants from control and AKO mouse iWAT and stimulated them with ACTH or Iso. As expected, Iso stimulated phosphorylation of PKA substrates to comparable extent in control and AKO iWAT explants (Fig. 7e). In contrast, ACTH-induced PKA activation was notably stronger in AKO mouse iWAT compared to control, indicating that BAF60a deficiency sensitizes response of iWAT to ACTH stimulation. In support of this, we found that induction of iWAT $Ucp1$ mRNA expression was significantly higher in AKO mice than control following ACTH treatment (Fig. 7f). Together, these studies demonstrate that BAF60a per se is not essential for cold-induced white fat browning; instead, its deficiency activates the ACTH-MC2R pituitary-adipose signaling axis to promote beige fat formation.

## Discussion

The brown and beige thermogenic programs are regulated by an overlapping set of transcription factors and cofactors. In this study, we uncover a surprisingly dichotomous role of BAF60a-mediated chromatin remodeling in thermogenic regulation (Fig. 7g). BAF60a is absolutely required for stimulation of brown fat thermogenesis in response to cold exposure and adrenergic activation; its deficiency impaired cold-induced thermogenesis and rendered mice hypersensitive to cold stress. At the molecular level, BAF60a physically associates with both cBAF- and BRD9-containing ncBAF complexes; the latter is required for thermogenic gene expression during brown adipogenesis and in response to adrenergic stimulation. BAF60a inactivation results in restricted chromatin accessibility near thermogenic genes such as $Ucp1$ in brown adipose tissue. Many of the ATAC-seq peaks regulated by BAF60a coincide with PPARγ and EBF2 binding sites, suggesting that these two transcription factors may direct BAF60a-mediated chromatin remodeling. PPARγ and EBF2 have been shown to recruit SWI/SNF complexes through inter-action with the histone demethylase JMJD1A and the ATPase BRG1, respectively[21,22]. Our results reveal a critical role of BAF60a in gating chromatin accessibility at these regulatory regions.

A surprising observation here is that BAF60a is essentially dispensable for induction of beige fat formation; its deficiency

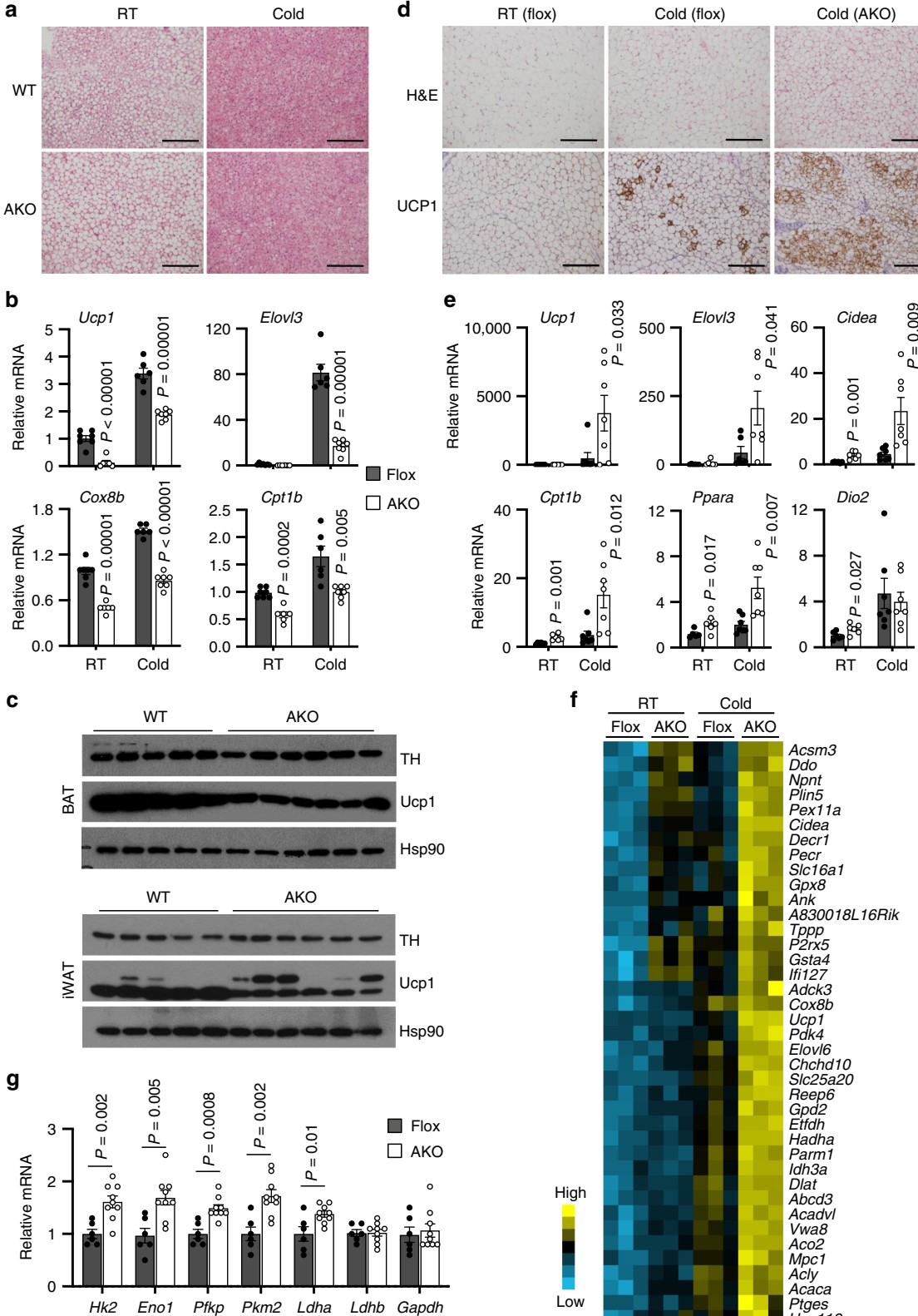

**Fig. 4 BAF60a deficiency promotes cold-induced beige fat formation. a** H&E staining of BAT from mice kept at ambient room temperature (RT) or after cold acclimation (Cold) (scale bar = 200 μm; $n = 4$ per group). **b** qPCR analysis of BAT gene expression (flox, $n = 7$ for RT, $n = 6$ for Cold; AKO, $n = 6$ for RT, $n = 7$ for Cold). **c** Immunoblots of brown fat and inguinal fat lysates. **d** H&E and UCP1 immunostaining of iWAT sections (scale bar = 200 μm) (flox, $n = 6$ for RT, $n = 7$ for Cold; AKO, $n = 8$ for Cold). **e** qPCR analyses of iWAT gene expression (flox, $n = 6$ for RT, $n = 6$ for Cold; AKO, $n = 6$ for RT, $n = 7$ for Cold). **f** Heatmap representation of microarray data showing a cluster of genes exhibiting elevated expression in AKO iWAT following cold acclimation. **g** qPCR analysis of genes enriched in glycolytic beige fat (flox, $n = 6$, filled; AKO, $n = 9$, open). Data in **b**, **e**, **g** represent mean ± s.e.m. two-tailed unpaired Student's $t$ test, flox vs. AKO. Source data are provided as a Source Data file.

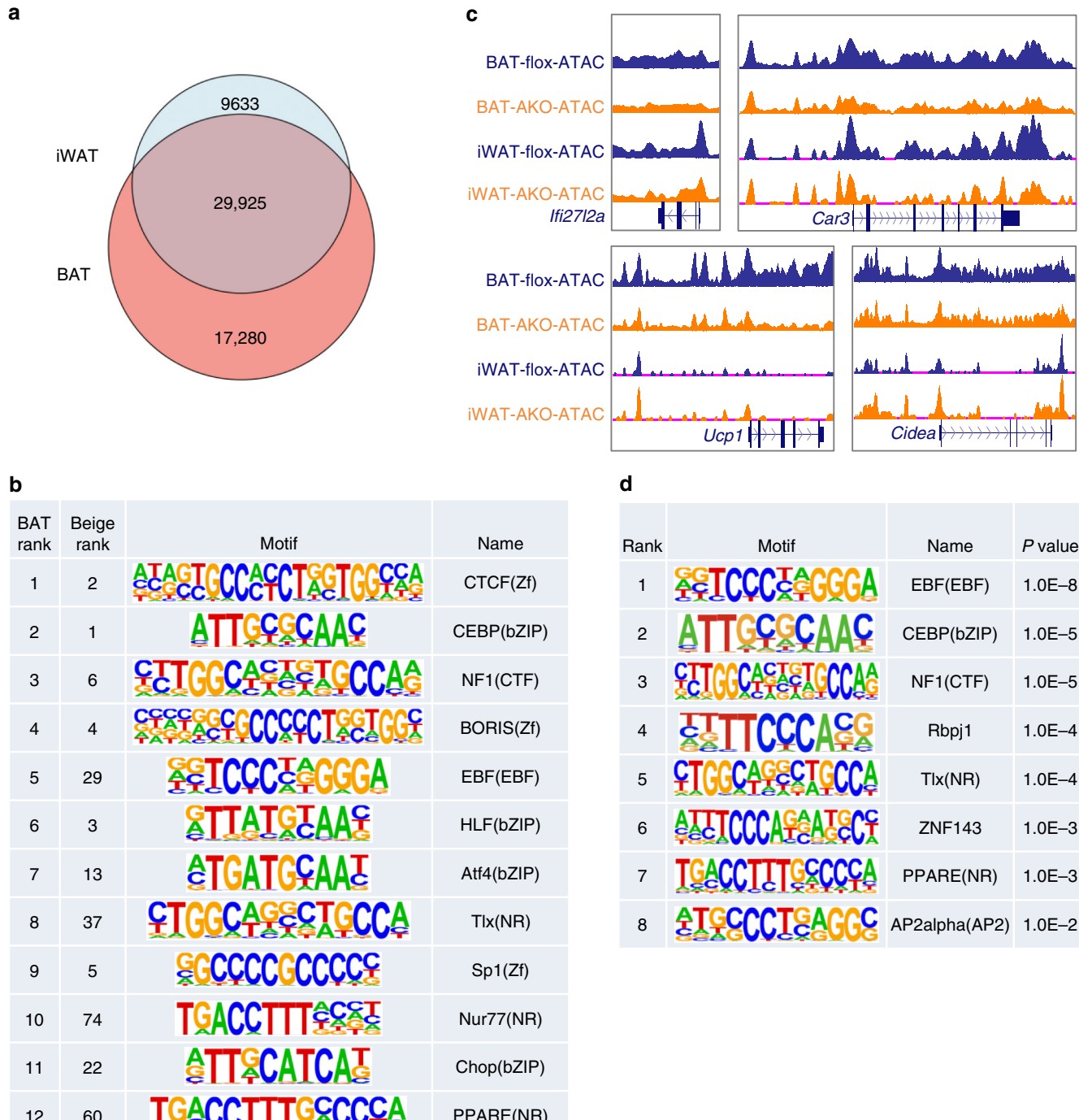

**Fig. 5 ATAC-seq analysis of iWAT from cold-acclimated mice. a** Overlap of ATAC-seq peaks detected in BAT and cold-acclimated iWAT. **b** Top 12 motifs enriched in BAT ATAC-seq peaks and their ranks in beige fat. **c** UCSC genome browser traces of ATAC-seq peaks at the *Ifi27l2a*, *Car3*, *Ucp1*, and *Cidea* loci. **d** Motif-enrichment analysis of the peaks upregulated in AKO iWAT.

instead augmented white fat browning following chronic cold acclimation. ATAC-seq studies indicate that BAF60a deficiency differentially affected chromatin accessibility in brown and beige fat. These results illustrate that, despite the similarity between the brown and beige thermogenic gene programs, BAF60a are differentially required for transcriptional activation of these genes. These findings are in contrast to other thermogenic regulators such as PRDM16 and EBF2, which are required for thermogenic gene activation in both brown and beige adipocytes[40–43]. Given that BAF60a is largely dispensable for cold-induced browning of white fat, it is possible that BAF60b and/or BAF60c may

compensate for the loss of BAF60a in beige adipocytes. Similar redundancy has been observed in a recent study on skeletal myocyte gene regulation by BAF60 family members[44]. Defects in brown fat thermogenesis have been shown to trigger compensatory browning response[45]. Our work here reveals a mechanism mediated by activation of the ACTH/MC2R pituitary-adipose endocrine pathway to promote iWAT browning. In this case, elevated MC2R expression sensitizes BAF60a-deficient adipocytes to PKA activation and thermogenic gene expression in response to ACTH. Recent studies have implicated ACTH and MRAP in thermogenic gene induction[38,46]. Our work illustrates that, in

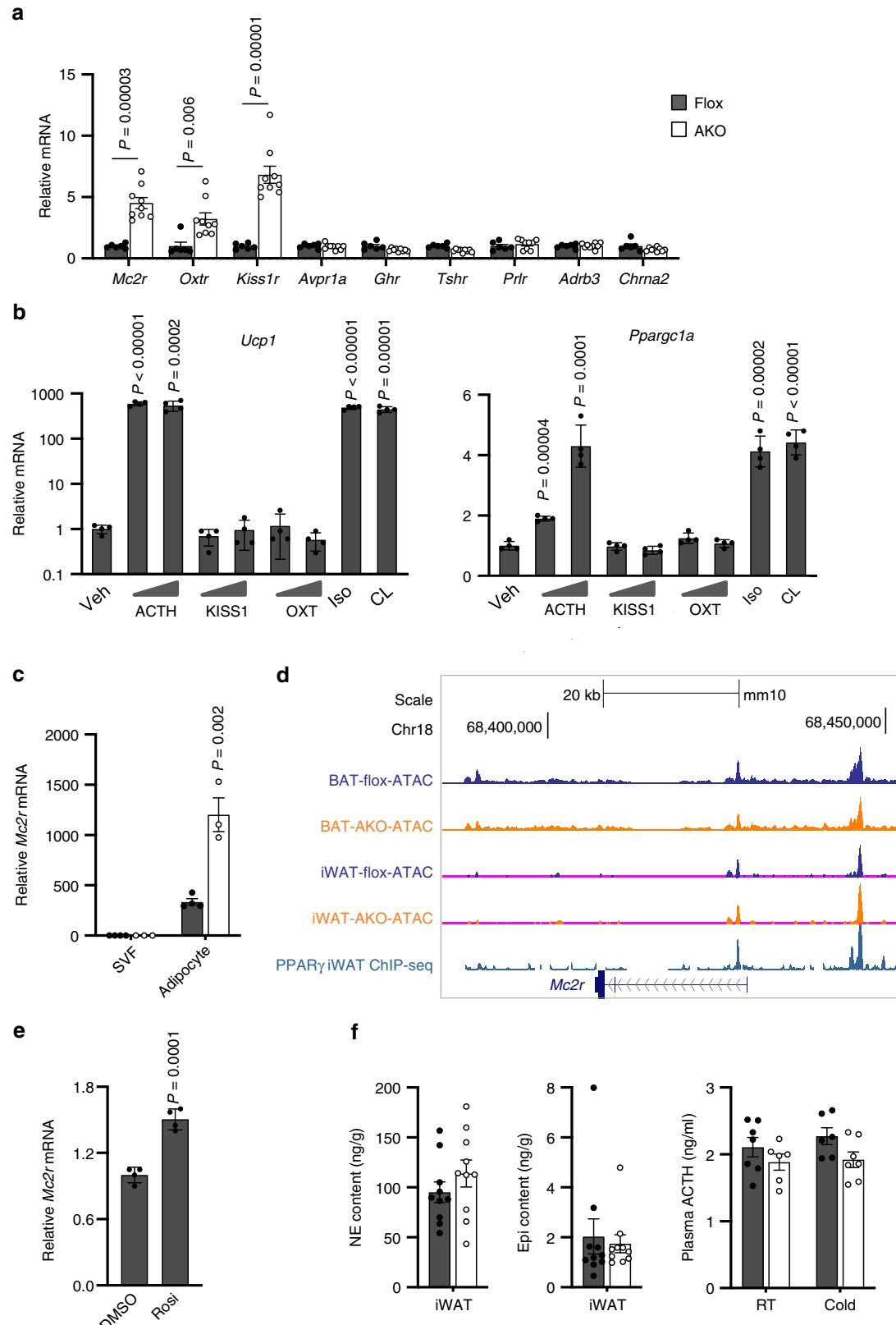

addition to the regulation of ACTH levels, the expression of MC2R modulates the sensitivity of adipocyte response to ACTH.

## Methods
**Animal studies**. All relevant ethical regulations for animal testing and research have been complied. All animal procedures were approved by the University of Michigan Institutional Animal Care and Use Committee. The protocol number is PRO00007676. Wild-type C57BL/6J mice (JAX stock #000664) were purchased from the Jackson Laboratory. BAF60a flox/flox mice were generated as previously described[25]. To inactivate BAF60a in adipocytes, BAF60a flox/flox mice were crossed with Adiponectin-Cre transgenic mice. Mice were fed ad lib with standard rodent chow and maintained under 12/12 h light/dark cycles. For cold exposure, 4-month-old mice were individually housed in cages pre-chilled at 4 °C. Mice had

**Fig. 6 Lack of BAF60a increases the expression of MC2R. a** qPCR analysis of iWAT gene expression in mice following cold acclimation (flox, $n = 6$, filled; AKO, $n = 9$, open). Data represent mean ± s.e.m.; two-tailed unpaired Student's $t$ test, flox vs. AKO. **b** qPCR analysis of gene expression in differentiated beige adipocytes after treatments (5 h) with Veh, ACTH, KISS1, or Oxytocin (OXT) at two doses ($n = 4$ per group). Two-tailed unpaired Student's $t$ test, treatments vs. Veh. **c** qPCR analysis of *Mc2r* expression in stromal vascular fraction (SVF) and mature adipocytes (Adipocyte) from iWAT of flox ($n = 4$, filled) and AKO ($n = 3$, open) mice. Data represent mean ± s.e.m. flox vs. AKO, two-tailed unpaired Student's $t$ test. **d** UCSC genome browser view of ATAC-seq peaks at the Mc2r locus. **e** qPCR analysis of *Mc2r* gene expression in beige adipocytes after 5 h treatments with DMSO or 1 μM rosiglitazone (Rosi) ($n = 4$ per group). Data represent mean ± s.d. Rosi vs. DMSO, two-tailed unpaired Student's $t$ test. **f** Norepinephrine (NE) and epinephrine (Epi) levels in iWAT after chronic cold acclimation (flox, $n = 10$, filled; AKO, $n = 10$, open). Plasma ACTH levels at ambient temperature (flox $n = 7$; AKO, $n = 6$) or following cold acclimation (flox $n = 6$; AKO $n = 7$). Data represent mean ± s.e.m. Source data are provided as a Source Data file.

free access to food and water throughout the experiment. A rectal thermometer was used to measure core body temperature. For cold acclimation, mice at 6–8 months of age were kept in a temperature-controlled chamber. Temperature in the chamber was lowered 3 °C per day until it reached 10 °C. Mice were then maintained at 10 °C for 7 more days before analysis. For ACTH treatment, ACTH (Phoenix Pharma) was prepared in phosphate-buffered saline (PBS) and injected along inguinal locations at a dose of 5.6 μg/kg body weight per side.

**Primary cell culture and differentiation.** Primary brown preadipocytes from BAF60a flox/flox pups were isolated and immortalized as previously described[47]. Interscapular brown adipose tissues were removed from postnatal day 1 pups. After mincing, the tissues were incubated for 20 min in Isolation buffer containing 1.5 mg/ml collagenase A (Roche) (123 mM NaCl, 5 mM KCl, 1.3 mM CaCl$_2$, 5 mM glucose, 100 mM HEPES, and 4% bovine serum albumin (BSA)) at 37 °C with constant rocking. The dissociated precursor cells were filtered through 70 μm strainer, followed by centrifugation. The cell pellets were resuspended in primary cell culture medium containing 20% fetal bovine serum/Dulbecco's modified Eagle's medium (FBS/DMEM) and incubated at 37 °C until reaching about 5% confluency. To immortalize the cells, retroviruses expressing SV40 large T antigen were used transduce the cells in the presence of 4 μg/ml polybrene. Two days later, the cells were selected in primary cell culture medium containing 0.45 mg/ml G418 until all the cells in control wells (cells without retroviral transduction) were dead. The subsequent immortalized brown preadipocytes were maintained in 10% FBS/DMEM. To delete BAF60a, preadipocytes were transduced with control or a retroviral vector expressing Cre recombinase. The resulting stable cell lines were maintained in DMEM supplemented with 10% FBS. To induce differentiation, confluent cultures were exposed to induction medium containing 10% FBS/DMEM supplemented with 0.5 μM IBMX, 125 μM indomethacin, 1 μM dexamethasone, 1 nM T3, and 20 nM insulin. After 48 h, the induction medium was replaced with maintenance medium (10% FBS/DMEM supplemented with 1 nM T3 and 20 nM insulin. Fresh maintenance medium was added every 2 days until day 7, when the cells were treated for 5 h with vehicle or 1 μM NE before harvest for RNA analyses. MitoTracker Orange (Thermo Fisher) staining was performed following the manufacturer's instruction. To study the function of BRD9 in brown adipocyte differentiation, BRD9 inhibitor dBRD9 (Tocris) was included in the culture medium (250 nM) throughout differentiation. Fresh inhibitor or dimethyl sulfoxide (DMSO) was added every 2 days. For NE stimulation, differentiated adipocytes were incubated with DMSO, I-BRD9 (Cayman), or dBRD9 for 6 h before the addition of saline or 1 μM NE for another 4 h (RNA) or 6 h (protein).

Primary cell isolation from inguinal fat and differentiation were performed, as described previously[48]. Inguinal adipose tissues were removed and pooled from five to seven mice. After extensive mincing, the tissues were incubated in 37 °C shaking water bath in the presence of 1.5 U/ml collagenase D and 2.4 U/ml dispase II (Roche) prepared in Hanks' balanced salt solution (HBSS) containing 10 mM CaCl$_2$. The collagenase digestion was stopped by the addition of 10% FBS in DMEM/F12 GlutaMAX, followed by 100 μm filtration to remove undigested debris. After centrifugation, cell pellets were resuspended and further clarified by filtering again through 40 μm cell strainers. Isolated cells were maintained in DMEM/F12 GlutaMAX supplemented with 15% FBS. To induce differentiation, confluent cells were incubated in the induction medium (DMEM/F12 GlutaMAX containing 10% FBS, 0.5 μg/ml insulin, 2 μM dexamethasone, 1 μM rosiglitazone, and 0.5 μM IBMX). Two or three days later, the medium was replaced with a maintenance medium (DMEM/F12 GlutaMAX supplemented with 10% FBS and 0.5 μg/ml insulin) for a total of 7–8 days when adipocytes were fully differentiated. The cells were harvested for RNA analysis following the 5 h treatment of saline and various hormones at increasing doses, ACTH (5 and 50 nM), KISS1 (112–121, 15, and 150 nM), and OXT (20 and 200 nM) (Phoenix Pharma), 1 μM Iso, and 1 μM CL-316,243. To overexpress or knockdown *Mc2r*, cells were exposed overnight to adeno-associated viruses (AAVs) expressing GFP or MC2R at $3.5 \times 10^{10}$ vg/well of 12-well plate, and scramble or shMc2r at $2 \times 10^{11}$ vg/well of 12-well plate together with 6 μg/ml polybrene. At 4–6 days after AAV infection, differentiated cells were treated with saline, ACTH, Iso, or forskolin (FSK) for 5 h before harvesting for RNA. For Western blot analyses, differentiated cells were starved in DMEM/F12 GlutaMAX containing BSA, followed by 15-min treatments of saline, ACTH, Iso, and FSK.

**Co-immunoprecipitation.** Nuclear extracts were prepared from differentiated brown adipocytes. After incubating in a hypotonic buffer (10 mM Tris (pH 7.5), 10 mM KCl, 1.5 mM MgCl$_2$), the nuclei were released from the cells by passing through 21-gauge needle 6–8 times, followed by centrifugation. The pellets were then resuspended in high salt buffer containing 20 mM Tris, pH 7.5, 0.42 M NaCl, 1.5 mM MgCl$_2$, 20% glycerol, 0.2 mM EDTA, and the protease inhibitors. Nuclear proteins were subsequently extracted by end-to-end rocking at 4 °C for 45 min before centrifugation at $16,000 \times g$ for 20 min. After measuring protein concentration using the Bradford method, 240 μg of nuclear extracts was incubated overnight with 2 μg of control immunoglobulin G (IgG) or mouse anti-BAF60a antibody before the addition of 20 μl Protein G agarose (Thermo Fisher) to capture antibody-bound protein complexes. After three washes, the immunoprecipitated proteins were eluted by boiling the beads in 1× sodium dodecyl sulfate (SDS) sample buffer.

**Histology and immunohistochemistry.** Adipose tissues were dissected and directly fixed overnight in 10% formalin before paraffin embedding and H&E (hematoxylin and eosin) staining. Immunohistochemical analysis of Ucp1 protein was performed as described before[49]. Briefly, unstained tissue sections were heated at 60 °C for 30 min before rehydration. After antigen retrieval by boiling the slides in 10 mM sodium citrate-citric acid solution (pH = 6.2) for 20 min, the tissue sections were incubated overnight at 4 °C with anti-Ucp1 antibody (UCP11-A, Alpha Diagnostic) prepared in blocking reagent (5% BSA, 0.5% Tween-20, 0.05% NaN$_3$ in PBS). The slides were then incubated with ImmPRESS (peroxidase) polymer anti-rabbit IgG reagent, followed by exposure to 3,3-diaminobenzidine (Vector Lab). The slides were subsequently dehydrated for mounting (Permount, Thermo Fisher). Images were captured on Olympus BX51 microscope.

**qPCR and immunoblotting analyses.** qPCR analysis of gene expression and immunoblotting was performed as previously described[49,50]. Total RNA was isolated from tissues and cultured adipocytes using PureLink RNA Isolation Kit (Thermo Fisher). qPCR primer information can be found in Supplementary Table 1. Total cell lysates were prepared in a buffer containing 50 mM Tris-HCl (pH = 7.8), 137 mM NaCl, 10 mM NaF, 1 mM EDTA, 1% Triton X-100, 10% glycerol, and the protease inhibitor cocktail (Roche), followed by three freeze/thaw cycles. Tissue lysates were prepared by homogenizing in a buffer containing 50 mM Tris (pH = 7.6), 130 mM NaCl, 5 mM NaF, 25 mM β-glycerophosphoate, 1 mM sodium orthovanadate, 10% glycerol, 1% Triton X-100, 1 mM dithiothreitol, 1 mM phenylmethanesulfonyl fluoride (PMSF), and the protease inhibitor cocktail. After centrifugation, tissue lysates were separated on SDS-polyacrylamide gel (SDS-PAGE) and analyzed by the antibodies below. Primary antibodies used were as follows: mouse anti-Baf60a (611728, BD Biosciences), rabbit anti-Baf60a (10998-2-AP, Proteintech), rabbit anti-UCP1 (UCP11-A, Alpha Diagnostic), rabbit anti-HSP90 (sc-7947, Santa Cruz), mouse anti-HSP90 (sc-13119, Santa Cruz), rabbit anti-phospho-PKA substrate (#9624, Cell Signaling), rabbit anti-BRD9 (#619368, Active Motif), rabbit anti-BRG1 (H-88, Santa Cruz), and rabbit anti-BAF47 (A301-087A, Bethyl). Detailed information can be found in Supplementary Table 2.

**Transmission electron microscopy.** Brown adipose tissues were quickly removed and placed in a Petri dish containing a fixative buffer (2.5% glutaraldehyde, 0.1 M phosphate buffer, pH 7.4). The tissues were sliced into small pieces and fixed for 24 h. Transmission electron microscopy (TEM) images were acquired using Philips CM-100 TEM at the University of Michigan Microscopy and Image Analysis Laboratory.

**ChIP assay.** ChIP analyses on brown adipose tissues were performed as previously described[21]. Briefly, brown fat tissue was minced and fixed in 1% formaldehyde for 15 min at room temperature, followed by quenching with 125 mM glycine for 5 min. The cross-linked tissue samples were further grounded using PowerGen 125 Homogenizer (Fisher Scientific). The pellets from crude lysates were resuspended in ChIP lysis buffer (0.6% SDS, 1% Triton X-100, 0.15 M NaCl, 1 mM EDTA, 20 mM Tris at pH 8, protease inhibitors and 1 mM PMSF), followed by sonication using probe sonicate in aluminum cooling block (Virtis Virsonic 100 Ultrasonic

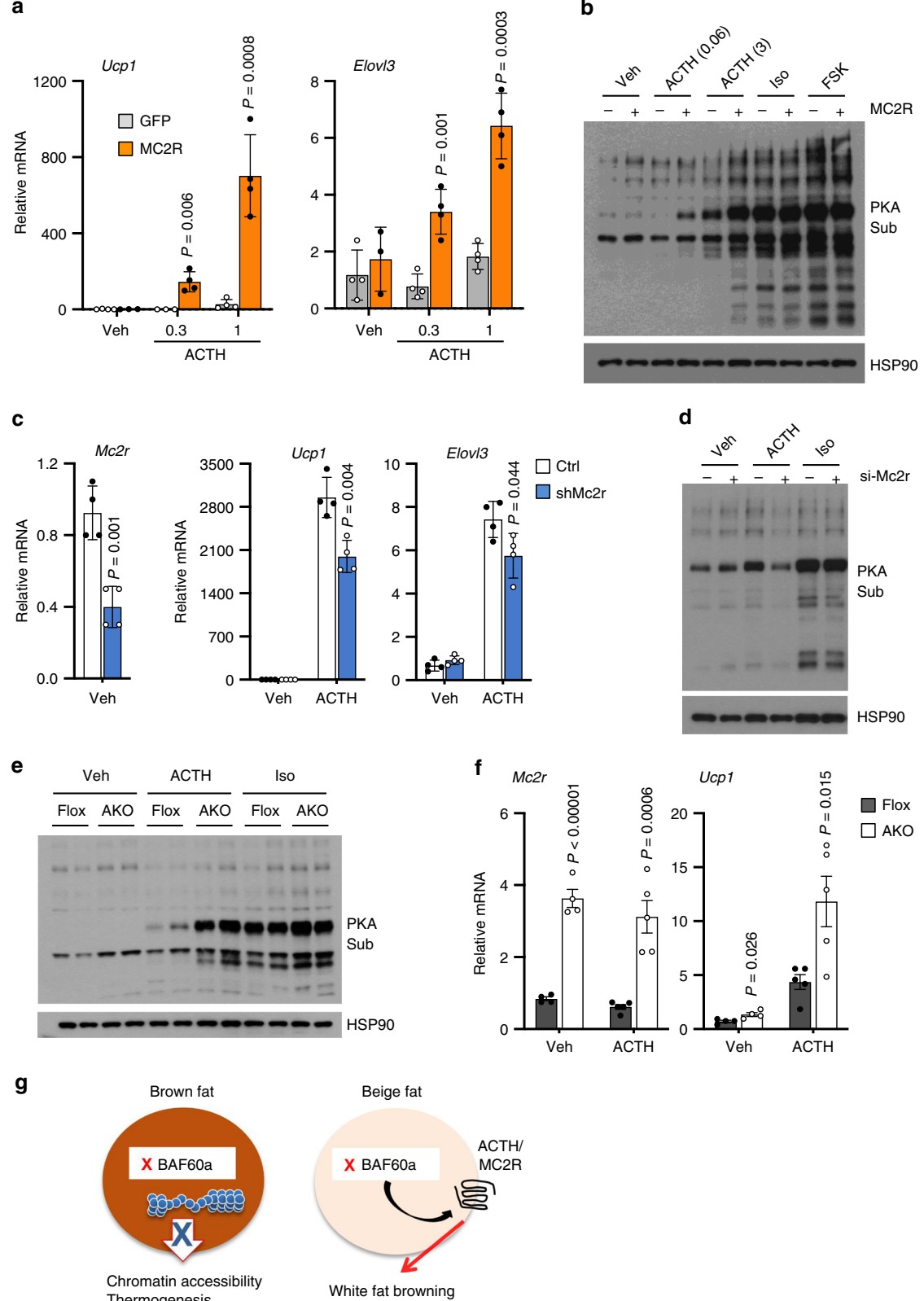

Cell Disrupter) for six times (10 s on, 30 s off) at set 6. After clearing by centrifugation, the chromatin lysates were diluted with ChIP dilution buffer (1% Triton X-100, 0.15 M NaCl, 1 mM EDTA, 20 mM Tris at pH 8) to a final concentration of 0.1% SDS and incubated overnight with antibodies at 4 °C with rotation. One microgram of antibodies against PPARγ (sc-7196x, Santa Cruz) or control IgG was used for each binding assay. To recover the antibody-bound chromatin, Protein A agarose beads (Thermo Fisher Scientific, #20334) were added

and incubated for 2 h at 4 °C. After washing with a series of wash buffers (A—0.1% SDS, 0.1% NaDOC, 1% Triton X-100, 0.15 M NaCl, 1 mM EDTA, 20 mM Tris at pH 8; B—0.1% SDS, 0.1% NaDOC, 1% Triton X-100, 0.5 M NaCl, 1 mM EDTA, 20 mM Tris at pH 8; C—0.25 M LiCl, 0.5% NaDOC, 0.5% NP-40, 1 mM EDTA, 20 mM Tris at pH 8; and D—1 mM EDTA, 20 mM Tris at pH 8), bound chromatin DNA was eluted in 1 mM EDTA/1% SDS buffer. Samples were reverse cross-linked overnight at 65 °C, followed by the treatment of RNase A and proteinase K. The

**Fig. 7 BAF60a deficiency triggers ACTH/MC2R signaling to promote browning. a** qPCR analysis of gene expression in beige adipocytes transduced with adeno-associated viruses (AAVs) expressing GFP (gray) or MC2R (orange). Cells were treated with 0, 0.3, and 1 nM ACTH for 5 h ($n = 4$ per group). Data represent mean ± s.d. ACTH vs. Veh, two-tailed unpaired Student's $t$ test. **b** Immunoblots of beige adipocytes transduced with AAV-GFP (−) or AAV-MC2R (+) following treatments with ACTH (0.06 or 3 nM), isoproterenol (Iso, 20 nM), or forskolin (FSK, 5 μM) for 15 min ($n = 2$ per group). **c** qPCR analysis of gene expression in beige adipocytes transduced with scramble (Ctrl, open) or shMc2r (blue) AAV vector. Differentiated adipocytes were treated with Veh or 10 nM ACTH for 5 h ($n = 4$ per group). Data represent mean ± s.d. shMc2r vs. Ctrl, two-tailed unpaired Student's $t$ test. **d** Immunoblots of transduced beige adipocytes treated with Veh, 3 nM ACTH or 20 nM Iso for 15 min ($n = 2$ per group). **e** Immunoblots of iWAT explant culture treated with Veh, 10 nM ACTH, or 30 nM Iso for 10 min. **f** qPCR analysis of iWAT gene expression from mice treated with Veh (flox, $n = 4$; AKO, $n = 4$) or ACTH (flox, $n = 5$; AKO, $n = 5$). Data represent mean ± s.e.m. flox (open) vs. AKO (filled); two-tailed unpaired Student's $t$ test. **g** A model depicting differential requirements for BAF60a in thermogenic gene expression in brown and beige fat. Source data are provided as a Source Data file.

---

DNA was further purified using PCR Purification Kit (Thermo Fisher). Quantitative real-time PCR using SYBR Green (Thermo Fisher) was performed using the primers located on Ucp1 promoters and *18s* gene. ChIP enrichment was normalized to input and IgG control.

**Infrared imaging analysis.** Infrared thermography was performed as previously described[51]. Briefly, mice were injected i.p. saline or 0.1 mg/kg body weight of CL-316,243 (Sigma-Aldrich) dissolved in saline. Three hours later, an infrared camera (T650sc, emissivity of 0.98, FLiR Systems) was used to capture the thermal pictures at a distance of ~30 cm above the cage. Multiple images were taken while mice could move freely. To analyze the thermal images, a region of interest (ROI) for each mouse was drawn for whole body (whole body) or the area above interscapular brown fat. The image files were then converted to CSV file using the FLiR Research IR program. Mean body surface temperature was calculated from top 10% pixels for each ROI.

**Cellular respiration assay.** Oxygen consumption rate was measured in fully differentiated brown adipocytes using Oxygen Meter (Strathkelvin Instruments) with a Mitocell (MT200) mixing chamber. Cells suspended in 400 μl maintenance medium (10% FBS/DMEM supplemented with 1 nM T3 and 20 nM insulin) were loaded into the mixing chamber. The change of oxygen concentration was recorded for 5 min, followed by injections of FCCP (carbonilcyanide *p*-triflouromethoxyphenylhydrazone, final 3 μM, Cayman) to the chamber. Oxygen consumption rate was calculated using software (782 Oxygen System version 4.0) and its sample was normalized to its protein content.

**Ex vivo culture and lipolysis assay.** After removing lymph nodes, inguinal adipose tissues collected from several mice were pooled and kept in 100 mm dishes containing 20 ml warm DMEM. Tissues were cut into small pieces (2–3 mm) using the scissors, followed by three PBS washes. Tissue pieces were then evenly distributed into 6-well tissue culture plates and maintained in 2 ml Medium 199 containing 1 nM insulin/T3. Three hours later, 1 nM ACTH (Phoenix Pharma), 30 nM Iso, or PBS were added and incubated for 15 min before harvest for protein analysis. For lipolysis assay, epididymal fat explants were obtained similarly as above. After 1 h incubation at 37 °C in HBSS containing 2% BSA, the tissues were stimulated with saline or 100 nM Iso for 3 h. NEFA concentrations in media were measured at different time points (Wako).

**Adipose ATAC-seq.** Brown adipose tissues were dissected from three mice for each genotype and snap-frozen individually in liquid nitrogen. For iWAT ATAC analyses, inguinal fat pads from cold-acclimated mice ($n = 6$ for each genotype) were harvested and pooled by genotype. The pooled inguinal adipose tissue was further minced and equally aliquoted prior to snap freezing.

Nuclei from brown and inguinal adipose tissue were isolated as previously described[52]. Five thousand nuclei were used in each transposase reaction (Ilumina), followed by barcoding and amplification of sequencing libraries[30]. DNA sequencing was performed on Illumina NextSeq-500 instrument with 38-bp paired-end reads for brown fat. HiSeq 4000 instrument with 150-bp paired-end reads was used for DNA sequencing in inguinal fat. The reads were trimmed to 38-bp paired-end reads by fastx_trimmer (http://hannonlab.cshl.edu/fastx_toolkit) for further processing. The raw and processed data were deposited to GEO (GSE128747). PPARγ and EBF2 ChIP-seq of brown fat bigwig files were downloaded from GEO database (GSE97116); PPARγ inguinal ChIP-seq bigwig file was downloaded from GEO sample GSM2433426. CrossMap (version 0.2.8) was used for liftover from mm9 to mm10. ChIP-seq peaks were called at false discovery rate (FDR) <0.05. The common peaks identified in ChIP-seq and ATAC-seq, and the common peaks of two adipose tissues in ATAC-seq were analyzed by Bedtools.

The pipeline of ATAC-seq data processing was adapted from a previous study[53]. Briefly, adapters were trimmed from paired-end reads using atactk (0.1.5), followed by mapping to the mm10 genome (bwa (0.7.17-r1188)). After marking duplicates with Picard Markduplicate (v.2.8.14; https://github.com/broadinstitute/picard), we filtered to autosomal and high-quality reads using samtools (v.1.7; samtools view -b -h -f 3 -F 4 -F 256 -F 1024 -F 2048 -q 30). For peak calling, macs2 (2.1.1.20160309) was used (--nomodel --shift -100 --extsize 200 -B --broad). Prior

to downstream analysis, the peaks were filtered against an mm10 blacklist (downloaded from http://mitra.stanford.edu/kundaje/akundaje/release/blacklists/mm10-mouse/mm10.blacklist.bed.gz). The treat_pileup.bedgraph files (generated by macs2) from each individual samples were normalized to its own library size. We took the average of treat_pileup.bedgraph files across samples within a biological condition to visualize ATAC-seq coverage in the UCSC genome browser. The final track files were displayed in the bigwig format, which was converted from normalized and averaged treat_pileup.bedgraph file by bedGraphToBigWig (v.4). Ataqv (v. 1.0.0; https://github.com/ParkerLab/ataqv) was used to perform the ATAC-seq data quality control analysis.

For downstream analysis in brown and inguinal adipose tissue, we took the union of all blacklist-filtered broad peaks and kept those overlapping with peak calls (FDR < 0.1) from at least two of three biological (brown) and technical (inguinal) replicates in either of the genotypes using bedtools' multiIntersectBed (v.2.27.1). Bedtools' coverageBed was used to count the reads in each peak for each sample. The motif-enrichment analysis was performed by HOMER (HOMER, findMotifs.pl) with different peak inputs.

The R package DESeq2 (v.1.20.0) was used to perform the differential peak analysis. The principle component 1 score of each sample was included as a covariate for brown adipose tissues. Differential peaks were determined with FDR < 0.1. The peaks with extremely low counts were excluded for further analysis (FDR = NA). To link the peak coordinates with their closest genes, peaks were annotated to genes within the (+1 kb, −100 bp) around the transcription start site by HOMER annotatePeaks.pl[54]. For transcription factor footprinting analysis, PPARγ and CTCF binding motifs were obtained from the MEME database. CENTIPEDE was used to call footprints as previously described[55]. The final footprint signal (cut point counts) were normalized to library size and averaged in each group.

**Statistical analysis.** Experiments of similar conditions were independently performed two or more times. Data were analyzed using Prism8 (GraphPad Software Inc.). Two-tailed unpaired Student's $t$ test and two-way ANOVA (analysis of variance) were used for statistical analysis as indicated in the figure legends. Statistical significance between groups was called when $P$ value is smaller than 0.05. Motif enrichment was statistically analyzed using cumulative binomial distribution (HOMER, findMotifs.pl).

**Reporting summary.** Further information on research design is available in the Nature Research Reporting Summary linked to this article.

## Data availability

The data that support the findings of this study are available from the corresponding authors upon reasonable request. ATAC-seq and microarray data have been deposited in GEO database (GSE128747 and GSE145498, respectively). The source data underlying Figs. 1a, b, d, e, h, 2d, 3a–c, e–j, 4b, c, e, g, 6a–c, e, f, and 7a–f and Supplementary Figs. 1, 2, 3a, 4, and 5 are provided as a Source Data file. PPARγ and EBF2 ChIP-seq of brown fat bigwig files were downloaded from GEO database (GSE97116); PPARγ inguinal ChIP-seq bigwig file was downloaded from GEO sample GSM2433426.

## Code availability

Code for processing and analyzing ATAC-seq data is available on GitHub (https://github.com/ltongyu/ATAC-seq-pipeline).

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

## Acknowledgements

This work was supported by NIH Grants (DK102456 and DK118731, J.D.L.), American Diabetes Association (#1-15-BS-118), Michigan Diabetes Research Center (DK020572), and Michigan Nutrition and Obesity Center (DK089503). We thank Xiaoling Peng for providing technical support.

## Author contributions

S.L. and J.D.L. conceived the project and designed research. S.L., T.L., L.M., J.X., Q.Y., L.Y., X.-Y.Z. and Z.-X.M. performed the experiments and analyzed data. T.L. analyzed ATAC-seq data with assistance from P.O. and S.C.J.P. S.L. and J.D.L. wrote the manuscript.

## Competing interests

The authors declare no competing interests.
