## [Peer Review File · Nature Communications]

Reviewers' comments:

Reviewer #1 (Remarks to the Author):

Here, Liu and colleagues show that BAF60a is required for brown fat thermogenesis upon cold challenge and pharmacological adrenergic activation. ATAC-seq reveals that adipocyte specific BAF60a deletion (AKO) results in very few sites of decreased accessibility. Among sites of decreased accessibility is the loci of the canonical thermogenesis regulator UCP1 which aligns with an observed decrease in UCP1 mRNA and protein. Additionally, decreased accessibility is observed at binding sites of key thermogenic regulator and mitochondrial biogenesis promoting transcription factors, PPAR γ and EBF2 (including binding sites at the UCP1 locus). At the organellar level, BAF60a AKO brown fat cells have fewer mitochondria number and correspondingly, lower maximal oxygen consumption, consistent with the observed inability to carry out thermogenesis.

Paradoxically, while UCP1 levels are decreased in BAF60a AKO brown fat, levels of UCP1 are increased in AKO inguinal white adipose tissue (iWAT). Furthermore, following cold challenge in iWAT, H&E staining & IHC reveal a beige fat phenotype and increased UCP1 levels, respectively. This in combination with microarray data showing increased transcription of genes associated with mitochondrial fuel oxidation and thermogenesis suggests browning of white fat. The underlying mechanism presented for white fat browning is upregulation of the MC2R nuclear receptor, sensitizing responses to its ligand ACTH in inducing transcription of thermogenesis regulators UCP1 and PGC-1 α . ACTH has previously been reported to activate brown fat thermogenesis and induce browning of white fat via this mechanism (van den Beukel, J.C., et al. FASEB 2014).

This study adds an interesting cell state transition (brown fat thermogenesis) to the list of cell state transitions dependent upon the BAF chromatin remodeling complex. The mechanism provided for impaired brown fat thermogenesis of both decreased accessibility at key thermogenic loci (UCP1) and binding sites of thermogenesis promoting TF binding sites (PPAR, EBF2) is compelling, although at remarkably very few sites (265). This observation at the chromatin level coupled with decreased transcript and protein levels of seminal thermogenesis effector UCP1, lower mitochondrial number, and decreased browning makes a case for decreased accessibility at key thermogenic loci as a potential mechanism for impaired brown fat thermogenesis in BAF60a AKO mice, although more biochemical and mechanistic insight is required here. While the paradoxical observation that iWAT undergoes browning in BAF60a AKO mice is interesting, the work on white fat browning lacks the mechanistic understanding seen with impairment of brown fat thermogenesis.

Major Points:

1. The authors should perform PPAR γ and EBF2 ChIP-seq in WT and BAF60A KO brown fat to determine if PPAR γ and EBF2 binding are affected BAF60A-dependent accessible sites in brown fat. A very small number of accessible sites are affected in BAF60A KOs relative to the number of PPAR γ and EBF2 sites (265 verses tens of thousands). Changes TF binding may reveal more extensive regulation by BAF60A at sites that do not necessarily show accessibility changes.

2. Recently, BAF60A was shown to be a component of the non-canonical BRD9-containing BAF complex (ncBAF), as well as the canonical BAF complex (Mashtalir et al. Cell 2018, Michel et al. NCB 2018, Alpsy et al. JBC 2016, Gatchalian et al. Nat Comm 2018). BAF60C, however, is specifically found in canonical BAF, but not ncBAF. The authors should thus profile BAF subunit expression by RNA and protein specifically looking for changes in BAF60 subunit expression (e.g. upregulation of BAF60B, BAF60C) in WT and BAF60A KO brown fat and iWAT. The authors should also profile BAF/ncBAF complex composition using BAF60A co-IP in brown fat and iWAT to determine if BAF60A is primarily associated with ncBAF or canonical BAF in these tissues.

3. If BAF60A is found to co-IP with BRD9, the authors should test whether the defects associated with BAF60A deletion in brown fat can be ascribed to ncBAF activity by treating adipocytes with I-BRD9 (inhibitor) or dBRD9 (degrader) and performing RT-qPCR for UCP1 and other targets.

4. In addition, BRD9 was shown to bind to an acetylated residue in the Vitamin D Receptor (Wei et al Cell 2018), which is conserved in other nuclear receptors, including RXR identified among BAF60A-dependent motifs shown in Figure 2C. I wonder whether the effects of BAF60A could be ascribed to a direct association of a BAF60A-containing ncBAF complex with PPARg/RXR. To address this point, the authors should perform co-IP with endogenous PPARg/RXR (not overexpression) and monitor the association of BAF60A and other ncBAF subunits, including BRD9.

3. The authors should perform ATAC-seq in iWAT in both WT and BAF60a AKO mice. Changes in accessibility and transcription factor binding motifs should provide unexplored mechanistic insight into the white fat browning phenotype in AKO mice.

4. Finally, the authors should perform genetic deletion experiments to demonstrate that the phenotype of iWAT browning following cold challenge is dependent upon enhanced sensitivity of MC2R for its ligand, ACTH in vivo. This may be tested by performing dose response experiments with ACTH via intraperitoneally injecting WT & BAF60a AKO mice with ACTH or vehicle then recording any differential changes in WT & AKO iWAT with histology via H&E and UCP1 levels via immunohistochemistry. In addition, the authors should perform knockdown of MC2R in BAF60A KO adipocytes and test the levels PKA substrate phosphorylation in response to ACTH and Ucp1, Ppargc1a upregulation to confirm dependence on MC2R.

Minor points:

1. Manuscript would benefit by briefly defining beige fat in the abstract or introduction.

2. It would be helpful to label data panels with the cell type that was used to acquire it. Multiple cell types are used in this study and transitions between them are sometimes unclear if only examining the figures (e.g., transitioning from iWAT to differentiated adipocytes in Figure 4). These transitions could be made clearer by specifying the cell type used to acquire data under figure panels (e.g.,

Differentiated Adipocytes underneath western blots figure 4E and iWAT underneath western blots in figure 4F).

3. In figure 4D, specify the compound being given to cells and its concentration (ACTH (nM)) below the Y axis, in line with dosages of ACTH stimulation of the X axis. Figure 4D groups samples by dosage but doesn't specify what is being dosed in the figure.

Reviewer #2 (Remarks to the Author):

The manuscript by Liu aimed to determine the role of BAF60a in the regulation of brown and beige fat development and thermogenesis. The authors addressed this question by generating adipocyte-specific knockout of BAF60a in mice and found that BAT thermogenesis was significantly reduced in Adipo-Baf60a KO mice. The authors further examined the underlying mechanism by performing ATAC-seq in BAT and showed that many of the dysregulated thermogenic genes are direct targets of PPAR γ , the master regulator of fat development. On the other hand, the authors found that Baf60a KO enhances beige fat biogenesis in the subcutaneous WAT. This unexpected increase in beige fat biogenesis is likely due to activation of MC2R, a receptor for ACTH, thereby sensitizing ACTH stimulation and enhancing PKA signaling.

These results together provided compelling evidence showing the opposing function of Baf60a in brown fat and beige fat development in vivo. The paper is well-written and the data are convincing. I would suggest the following points to strengthen the authors' conclusion further.

1. The authors conclude that BAF60a is differentially required for brown fat vs. beige fat development. However, an alternative possibility to interpret the data would be that increased beige fat in Adipo-Baf60a is a compensatory activation of impaired BAT activation in vivo. The authors wish to address this by testing if the increased beige fat is observed under a thermoneutral condition, and also this activation, such as MC2R activation, is seen in a cell-autonomous fashion.

2. The mechanism by which MC2R expression is upregulated in the inguinal WAT remains less clear. Is MC2R a direct target of Baf60a and PPAR γ ? The authors also wish to comment on how the induction in MC2R is selective to the inguinal WAT (or not).

3. The finding that a large number of BAF60a target genes are co-localized with PPAR γ sites is intriguing. However, what is less unclear here is how the BAF60a recruitment regulated by cold stimuli. Is the PPAR γ -BAF60 interaction regulated by norepinephrine or cAMP signaling? The authors should also confirm the PPAR γ -BAF60 interaction using endogenous proteins.

4. The authors found enhanced PKA signaling in the WAT of Adipo-Baf60a KO mice. It is possible that increased lipolysis in the inguinal and visceral WAT may contribute to increased plasma TAG level. The authors should test if KO mice have higher lipolysis in the WAT.

Reviewer #1

Here, Liu and colleagues show that BAF60a is required for brown fat thermogenesis upon cold challenge and pharmacological adrenergic activation. ATAC-seq reveals that adipocyte specific BAF60a deletion (AKO) results in very few sites of decreased accessibility. Among sites of decreased accessibility is the loci of the canonical thermogenesis regulator UCP1 which aligns with an observed decrease in UCP1 mRNA and protein. Additionally, decreased accessibility is observed at binding sites of key thermogenic regulator and mitochondrial biogenesis promoting transcription factors, PPAR γ and EBF2 (including binding sites at the UCP1 locus). At the organellar level, BAF60a AKO brown fat cells have fewer mitochondria number and correspondingly, lower maximal oxygen consumption, consistent with the observed inability to carry out thermogenesis.

Paradoxically, while UCP1 levels are decreased in BAF60a AKO brown fat, levels of UCP1 are increased in AKO inguinal white adipose tissue (iWAT). Furthermore, following cold challenge in iWAT, H&E staining & IHC reveal a beige fat phenotype and increased UCP1 levels, respectively. This in combination with microarray data showing increased transcription of genes associated with mitochondrial fuel oxidation and thermogenesis suggests browning of white fat. The underlying mechanism presented for white fat browning is upregulation of the MC2R nuclear receptor, sensitizing responses to its ligand ACTH in inducing transcription of thermogenesis regulators UCP1 and PGC-1 α . ACTH has previously been reported to activate brown fat thermogenesis and induce browning of white fat via this mechanism (van den Beukel, J.C., et al. FASEB 2014).

This study adds an interesting cell state transition (brown fat thermogenesis) to the list of cell state transitions dependent upon the BAF chromatin remodeling complex. The mechanism provided for impaired brown fat thermogenesis of both decreased accessibility at key thermogenic loci (UCP1) and binding sites of thermogenesis promoting TF binding sites (PPAR, EBF2) is compelling, although at remarkably very few sites (265). This observation at the chromatin level coupled with decreased transcript and protein levels of seminal thermogenesis effector UCP1, lower mitochondrial number, and decreased browning makes a case for decreased accessibility at key thermogenic loci as a potential mechanism for impaired brown fat thermogenesis in BAF60a AKO mice, although more biochemical and mechanistic insight is required here. While the paradoxical observation that iWAT undergoes browning in BAF60a AKO mice is interesting, the work on white fat browning lacks the mechanistic understanding seen with impairment of brown fat thermogenesis.

We thank this reviewer for his/her enthusiasm and insightful suggestions. We have performed a number of new studies to address these comments, as described below. Briefly, our new studies demonstrate that BAF60a is present in both cBAF and ncBAF remodeling complexes. The ncBAF subunit BRD9 is required for full induction of thermogenic genes during brown adipogenesis and in response to adrenergic stimulation. In addition, we performed ATAC-seq studies on WT and BAF60a AKO iWAT and showed that BAF60a deficiency elicits opposing effects on thermogenic gene loci in brown and beige fat. Finally, we obtained additional data to support our conclusion that MC2R levels dictate beige adipocyte responsiveness to ACTH stimulation. These results greatly strengthen our conclusions and have been incorporated in the revised manuscript.

Major Points:

1. The authors should perform PPAR γ and EBF2 ChIP-seq in WT and BAF60A KO brown fat to determine if PPAR γ and EBF2 binding are affected BAF60A-dependent accessible sites in brown fat. A very small number of accessible sites are affected in BAF60A KOs

relative to the number of PPAR γ and EBF2 sites (265 verses tens of thousands). Changes TF binding may reveal more extensive regulation by BAF60A at sites that do not necessarily show accessibility changes.

As accurately noted by this reviewer, we only observed a small number of BAF60a-dependent ATAC-seq peaks. Previous studies have demonstrated that BRG1 deficiency does not alter EBF2 occupancy at its cis-regulatory elements, suggesting that transcription factor binding precedes chromatin remodeling and changes in accessibility (*Shapira et al. 2017, Genes Dev. 31,1-14*). In our study, we found that BAF60a deficiency markedly reduces the expression of PPAR γ 2 in brown fat (**Figure 1D**). As such, decreased PPAR γ levels will confound the interpretation of ChIP-seq data and make it impossible to attribute any observed changes to altered occupancy vs. reduced expression. Nevertheless, we appreciate this reviewer's comment and agree that it is likely that chromatin accessibility and TF occupancy may not be always co-regulated at the genomic level.

2. Recently, BAF60A was shown to be a component of the non-canonical BRD9-containing BAF complex (ncBAF), as well as the canonical BAF complex (Mashtalir et al. Cell 2018, Michel et al. NCB 2018, Alpsy et al. JBC 2016, Gatchalian et al. Nat Comm 2018). BAF60C, however, is specifically found in canonical BAF, but not ncBAF. The authors should thus profile BAF subunit expression by RNA and protein specifically looking for changes in BAF60 subunit expression (e.g. upregulation of BAF60B, BAF60C) in WT and BAF60A KO brown fat and iWAT. The authors should also profile BAF/ncBAF complex composition using BAF60A co-IP in brown fat and iWAT to determine if BAF60A is primarily associated with ncBAF or canonical BAF in these tissues.

We have profiled the expression of other BAF subunits in WT and BAF60a AKO BAT, iWAT, and eWAT. While BAF60b and BAF60c mRNA levels remained unaltered, expression of the ncBAF subunit Brd9 was increased by approximately two-fold in BAF60a-deficient brown fat (**Figure 1D and Supplementary Figure 1B**). We performed BAF60a co-IP studies on brown adipocyte lysates and demonstrated that BAF60a physically associates with endogenous BAF47 and BRD9 (**new Figure 3F**). These results indicate that BAF60a is present in both cBAF and ncBAF complexes in brown adipocytes.

3. If BAF60A is found to co-IP with BRD9, the authors should test whether the defects associated with BAF60A deletion in brown fat can be ascribed to ncBAF activity by treating adipocytes with I-BRD9 (inhibitor) or dBRD9 (degrader) and performing RT-qPCR for UCP1 and other targets.

This is an excellent suggestion. We performed treatments on differentiating brown adipocytes with BRD9 inhibitor (I-BRD9) and a BRD9 degrader (dBRD9). Very interestingly, BRD9 inhibition strongly diminished the induction of UCP1 mRNA and protein expression during brown adipogenesis (**new Figure 3G-H**). In addition, the induction of UCP1 in differentiated brown adipocytes in response to adrenergic stimulation was also greatly impaired by inhibition of BRD9 (**new Figure 3I-J**). These results underscore a previously unappreciated role of ncBAF in the control of thermogenic gene program. Future studies using genetic tools are needed to delineate the physiological significance of ncBAF in adipose thermogenesis, metabolism, and systemic physiology.

4. In addition, BRD9 was shown to bind to an acetylated residue in the Vitamin D Receptor (Wei et al Cell 2018), which is conserved in other nuclear receptors, including RXR identified among BAF60A-dependent motifs shown in Figure 2C. I wonder whether the effects of BAF60A could be ascribed to a direct association of a BAF60A-containing ncBAF complex with PPAR γ /RXR. To address this point, the authors should perform co-IP with endogenous PPAR γ /RXR (not overexpression) and monitor the association of BAF60A and other ncBAF

subunits, including BRD9.

We appreciate this reviewer's point. However, we were unable to detect the formation of stable transcriptional complexes containing endogenous PPAR γ and BAF complexes in co-IP studies. It is possible that the recruitment of BAF complexes to PPAR γ is dynamic and transient in nature, unlike the core subunits within the BAF complexes.

3. The authors should perform ATAC-seq in iWAT in both WT and BAF60a AKO mice. Changes in accessibility and transcription factor binding motifs should provide unexplored mechanistic insight into the white fat browning phenotype in AKO mice.

Per reviewer's suggestion, we have performed ATAC-seq on iWAT from WT and BAF60a AKO mice following cold acclimation. We observed extensive overlap of ATAC-seq peaks between BAT and iWAT, suggesting that brown fat and beige fat share highly similar chromatin accessibility landscape (**new Figure 5**). While chromatin accessibility for a subset of genes (e.g. *Ifi2712a* and *Car3*) was diminished in both fat depots, BAF60a deficiency resulted in opposing effects on chromatin accessibility for thermogenic genes such as *Ucp1* and *Cidea*. These results provide further support for our conclusion that BAF60a is differentially required for thermogenic gene expression in brown fat and beige fat.

4. Finally, the authors should perform genetic deletion experiments to demonstrate that the phenotype of iWAT browning following cold challenge is dependent upon enhanced sensitivity of MC2R for its ligand, ACTH in vivo. This may be tested by performing dose response experiments with ACTH via intraperitoneally injecting WT & BAF60a AKO mice with ACTH or vehicle then recording any differential changes in WT & AKO iWAT with histology via H&E and UCP1 levels via immunohistochemistry. In addition, the authors should perform knockdown of MC2R in BAF60A KO adipocytes and test the levels PKA substrate phosphorylation in response to ACTH and *Ucp1*, *Ppargc1a* upregulation to confirm dependence on MC2R.

We appreciate the reviewer's point. However, the plasma half-life of ACTH is extremely short, making in vivo chronic studies on iWAT browning technically challenging. In addition, systemic administration of ACTH will activate the adrenal stress response pathway and confound the studies on adipose thermogenesis. As such, we performed acute treatment studies and demonstrated that ACTH responsiveness was enhanced in BAF60a-deficient iWAT (**Figure 7E-F**). In addition, we performed ACTH treatments in MC2R overexpression and knockdown beige adipocytes and showed that MC2R expression levels are causally linked to ACTH response (PKA activation and thermogenic gene expression) (**new Figure 7A-D**). Together, our data support the notion that enhanced ACTH/MC2R signaling contributed to increased browning in BAF60a AKO mice. Future studies using fat-specific *Mc2r* knockout mice will provide a critical test for the importance of this pathway in physiological regulation of adaptive thermogenesis and energy balance.

Minor points:

1. Manuscript would benefit by briefly defining beige fat in the abstract or introduction. We have added a sentence describing beige adipocytes in introduction.

2. It would be helpful to label data panels with the cell type that was used to acquire it. Multiple cell types are used in this study and transitions between them are sometimes unclear if only examining the figures (e.g., transitioning from iWAT to differentiated adipocytes in Figure 4). These transitions could be made clearer by specifying the cell type used to acquire data under figure panels (e.g., Differentiated Adipocytes underneath western blots figure 4E and iWAT underneath western blots in figure 4F).

We have incorporated the cell type details in relevant figure legends.

3. In figure 4D, specify the compound being given to cells and its concentration (ACTH (nM)) below the Y axis, in line with dosages of ACTH stimulation of the X axis. Figure 4D groups samples by dosage but doesn't specify what is being dosed in the figure.

We have updated the figures and the legends to include this information.

Reviewer #2

The manuscript by Liu aimed to determine the role of BAF60a in the regulation of brown and beige fat development and thermogenesis. The authors addressed this question by generating adipocyte-specific knockout of BAF60a in mice and found that BAT thermogenesis was significantly reduced in Adipo-Baf60a KO mice. The authors further examined the underlying mechanism by performing ATAC-seq in BAT and showed that many of the dysregulated thermogenic genes are direct targets of PPAR γ , the master regulator of fat development. On the other hand, the authors found that Baf60a KO enhances beige fat biogenesis in the subcutaneous WAT. This unexpected increase in beige fat biogenesis is likely due to activation of MC2R, a receptor for ACTH, thereby sensitizing ACTH stimulation and enhancing PKA signaling.

These results together provided compelling evidence showing the opposing function of Baf60a in brown fat and beige fat development in vivo. The paper is well-written and the data are convincing. I would suggest the following points to strengthen the authors' conclusion further.

We appreciate this reviewer's enthusiasm and helpful comments. We have performed several new studies to address these points, as described below.

1. The authors conclude that BAF60a is differentially required for brown fat vs. beige fat development. However, an alternative possibility to interpret the data would be that increased beige fat in Adipo-Baf60a is a compensatory activation of impaired BAT activation in vivo. The authors wish to address this by testing if the increased beige fat is observed under a thermoneutral condition, and also this activation, such as MC2R activation, is seen in a cell-autonomous fashion.

We agree with this reviewer that impaired brown fat thermogenesis likely triggers compensatory responses in BAF60a AKO mice. In this context, we showed that BAF60a is absolutely required for thermogenic gene expression in brown fat. In contrast, increased iWAT browning was observed in AKO mice following cold acclimation, suggesting that physiological compensatory signals are sufficient to override the absence of BAF60a and promote beige fat formation. We have performed ACTH treatments in MC2R overexpression and knockdown beige adipocytes and showed that MC2R expression levels are causally linked to ACTH response (PKA activation and thermogenic gene expression) (**new Figure 7A-D**). Interestingly, while there is no increased beige fat formation in AKO mice housed at thermoneutral temperature, Mc2r expression remained elevated in iWAT from Baf60a deficient mice (**new Supplementary Figure 5C**), suggesting that BAF60a likely exerts a cell-autonomous effect on Mc2r expression.

2. The mechanism by which MC2R expression is upregulated in the inguinal WAT remains less clear. Is MC2R a direct target of Baf60a and PPAR γ ? The authors also wish to comment on how the induction in MC2R is selective to the inguinal WAT (or not).

We added a sentence (page 9) to indicate that Mc2r induction was also observed in BAF60a AKO mouse brown and epididymal fat pads. Regarding the mechanisms that regulate Mc2r expression, we explored chromatin features near the Mc2r gene locus and identified two PPAR γ binding sites (ChIP-seq) that correspond to two prominent ATAC-seq peaks (**new Figure 5D**). In addition, rosiglitazone treatment increased Mc2r mRNA expression in

cultured beige adipocytes (**new Figure 5E**), suggesting that augmented PPAR γ signaling may contribute to Mc2r induction in adipocytes lacking BAF60a.

3. The finding that a large number of BAF60a target genes are co-localized with PPAR γ sites is intriguing. However, what is less unclear here is how the BAF60a recruitment regulated by cold stimuli. Is the PPAR γ -BAF60 interaction regulated by norepinephrine or cAMP signaling? The authors should also confirm the PPAR γ -BAF60 interaction using endogenous proteins.

This is an interesting point. However, we were unable to detect the formation of stable endogenous transcriptional complexes containing PPAR γ and BAF complexes in co-IP studies. It is possible that the recruitment of BAF complexes to PPAR γ is dynamic and transient in nature, unlike physical association among the core subunits within the BAF complexes.

4. The authors found enhanced PKA signaling in the WAT of Adipo-Baf60a KO mice. It is possible that increased lipolysis in the inguinal and visceral WAT may contribute to increased plasma TAG level. The authors should test if KO mice have higher lipolysis in the WAT.

We performed in vitro lipolysis assay on WT and BAF60a AKO eWAT and observed increased rate of lipolysis under basal and stimulated conditions (**new Supplementary Figure 1A**). These results suggest that enhanced lipolysis may contribute to elevated plasma lipid levels. It is likely that defects in lipid uptake and oxidation by brown fat may also contribute to the plasma lipid phenotype. In fact, the BAF60a AKO mice were exquisitely sensitive to cold exposure and unable to defend their core body temperature.

REVIEWERS' COMMENTS:

Reviewer #1 (Remarks to the Author):

The authors have addressed my comments. I applaud them on a very nice study.

Reviewer #2 (Remarks to the Author):

The authors addressed the reviewer's comments satisfactory. I have no more concerns.

Reviewer #1 (Remarks to the Author):

The authors have addressed my comments. I applaud them on a very nice study.

We thank this reviewer for his/her enthusiasm and are grateful for acceptance of our work.

Reviewer #2 (Remarks to the Author):

The authors addressed the reviewer's comments satisfactory. I have no more concerns.

We are delightful to be able to address the reviewer's comments and grateful for his/her acceptance of our study.